# Rhizospheric Communication through Mobile Genetic Element Transfers for the Regulation of Microbe–Plant Interactions

**DOI:** 10.3390/biology10060477

**Published:** 2021-05-28

**Authors:** Yee-Shan Ku, Zhili Wang, Shaowei Duan, Hon-Ming Lam

**Affiliations:** Centre for Soybean Research of the State Key Laboratory of Agrobiotechnology and School of Life Sciences, The Chinese University of Hong Kong, Hong Kong, China; ysamyku@cuhk.edu.hk (Y.-S.K.); wangzhili0804@gmail.com (Z.W.); duanshaowei515@gmail.com (S.D.)

**Keywords:** mobile genetic element (MGE), rhizosphere, soil microbe, microbe–plant interaction, symbiosis, bioremediation, horizontal gene transfer (HGT)

## Abstract

**Simple Summary:**

Rhizosphere, where microbes and plants coexist, is a hotspot of mobile genetic element (MGE) transfers. It was suggested that ancient MGE transfers drove the evolution of both microbes and plants. On the other hand, recurrent MGE transfers regulate microbe-plant interaction and the adaptation of microbes and plants to the environment. The studies of MGE transfers in the rhizosphere provide useful information for the research on pathogenic/ beneficial microbe-plant interaction. In addition, MGE transfers between microbes and the influence by plant root exudates on such transfers provide useful information for the research on bioremediation.

**Abstract:**

The transfer of mobile genetic elements (MGEs) has been known as a strategy adopted by organisms for survival and adaptation to the environment. The rhizosphere, where microbes and plants coexist, is a hotspot of MGE transfers. In this review, we discuss the classic mechanisms as well as novel mechanisms of MGE transfers in the rhizosphere. Both intra-kingdom and cross-kingdom MGE transfers will be addressed. MGE transfers could be ancient events which drove evolution or recurrent events which regulate adaptations. Recent findings on MGE transfers between plant and its interacting microbes suggest gene regulations brought forth by such transfers for symbiosis or defense mechanisms. In the natural environment, factors such as temperature and soil composition constantly influence the interactions among different parties in the rhizosphere. In this review, we will also address the effects of various environmental factors on MGE transfers in the rhizosphere. Besides environmental factors, plant root exudates also play a role in the regulation of MGE transfer among microbes in the rhizosphere. The potential use of microbes and plants for bioremediation will be discussed.

## 1. Introduction

Microbes, such as bacteria, fungi and oomycetes, and plants coexist in the rhizosphere. To adapt to each other’s presence and the constantly changing environment, there are extensive communications among all parties involved. The communication among microbes and plants in the rhizosphere by signaling molecules has been reviewed [1,2]. Examples of these signaling molecules include phytohormone-like molecules, antimicrobial molecules, organic acids and flavonoids [1,2]. The interaction between plants and rhizospheric microbes could also be regulated by secretory peptides [1,3]. In addition to these signaling molecules and peptides, there are an increasing number of reports showing evidence of the transfer of mobile genetic elements (MGEs), including DNA and RNA, between organisms in the rhizosphere. MGE transfers can occur within the same kingdom, such as between bacteria or between fungi, as well as across kingdoms, including those between microbes and plants. The transfers of MGEs in the rhizosphere have been shown to be ancient events which drove the evolution of species to adapt to their surrounding environments, and they can also be recurrent events that are still ongoing, regulating the interactions among different parties in the rhizosphere. The transfer of MGEs between bacteria for degrading toxic materials in the soil has been known as a survival strategy for the bacteria. These transfers are influenced by environmental factors such as temperature and soil composition. Nevertheless, it has been reported that plant root exudates mediate such transfers. Different plant species were found to have different effects on the MGE transfer among bacteria. The understanding of MGE transfers in the rhizosphere will further reveal the mechanisms of regulation of microbe–plant interactions, for both beneficial symbiosis and possible strategies for soil bioremediation.

## 2. General Introduction to Classic MGE Transfers

Mobile genetic elements (MGEs) are genetic materials that possess intracellular or intercellular mobility. Intracellular mobility refers to the movement within the genome of a cell, while intercellular mobility refers to the transfer from one species or cell to another [4]. In this review, the intercellular movement of MGEs between different species will be focused on. Typical examples of MGEs include plasmids, transposons, tiny RNAs and prophages [4,5,6]. Prophage refers to the bacteriophage DNA that exists as a plasmid in the host bacterium or integrates into the chromosome of the host bacterium [7,8,9]. Recently, it was also shown that phage DNA could integrate to the plasmid in the bacterial host to form a phage–plasmid complex [10]. The presence of prophages can protect the host bacteria from superinfection by closely related bacteriophages and also allows the hosts to colonize or survive in new ecological niches [11].

### 2.1. MGE Transfer in Prokaryotes

Horizontal transfer of MGEs in prokaryotes is common. Transformation, transduction, and conjugation are regarded as the classic MGE transfer mechanisms in prokaryotes [4]. Briefly, transformation involves the transfer of naked DNA, including chromosome-derived DNA and plasmid, between bacteria [4,12]. Conjugation requires physical contact between the cells. Conjugation involves conjugative plasmids to mediate their own transfer from the donor cell to the adjacent recipient cell through a protein structure known as pilus. Transduction refers to the horizontal gene transfer mediated by phage [13].

### 2.2. MGE Transfers in Eukaryotes

Phagotrophy and anastomosis are the two classic and major HGT mechanisms in eukaryotes. Phagotrophic eukaryotes engulf prey cells to enable horizontal gene transfer [14]. Phagotrophy has been suggested to be a major means of gene transfer into eukaryotes [15]. For instance, it was hypothesized that photosynthetic eukaryotes evolved as a result of the endosymbiosis of a photosynthetic cyanobacterium in phagotrophic eukaryotes which were non-photosynthetic [16]. The endosymbiosis enabled the subsequent engulfment of the prokaryotic cells by the eukaryotic host [14]. Fungal species frequently exchange genetic materials through specialized hyphae [17,18]. Such process is named anastomosis [17,18].

### 2.3. MGE Transfer between Microbes and Plants

HGT from prokaryotes to plants is common. One of the well-known examples of HGT from prokaryotes to plant is the transfer of tumor-inducing genes from *Agrobacterium* to the genome of the host plant [19]. Briefly, the transcription of *vir* genes in *Agrobacteria* is induced by phenolic compounds released by wounded plants. *Vir* genes are essential for producing and transporting single-stranded copies of the DNA fragment to be transferred. After passing through cell walls and cell membranes, the DNA fragment enters the nucleus of the host plant cell and integrates into the genome [20,21]. Recently, MGE transfers from plants to microbes have also been reported [22,23,24,25]. The mechanisms of such atypical MGE transfers, which are supported by phylogenetic analysis or experimental data, will be discussed in Section 3.3.3.

MGEs exist in most of the sequenced species genomes. The rhizosphere is a hotspot of communications among multiple species, including bacteria, fungi, and plants. Many studies have demonstrated that MGEs act as important media for such communications. Classic MGE transfers usually involve DNA. Recently, it has been reported that MGEs could also be RNAs in nature. Specific examples of MGE transfers through classic or novel mechanisms in the rhizosphere will be explored in Section 3.

## 3. MGE Transfers in the Rhizosphere

In the rhizosphere, microbes including viruses, bacteria, fungi, and oomycetes interact with the roots of plants [26]. MGEs play an important role in the regulation of interactions among different parties in the rhizosphere.

### 3.1. MGE Transfers among Soil Bacteria

Horizontal gene transfer (HGT) among bacteria is a widely known mechanism of MGE transfer in the rhizosphere [27,28]. Among soil bacteria, HGT is a common strategy to spread adaptation-related genes such as those related to antibiotic resistance and heavy metal resistance [29,30,31]. Soil contamination by antibiotics and heavy metals is becoming more widespread as a result of increased anthropogenic activities [32]. The spread of antibiotic and heavy metal resistance genes is important for bacteria to adapt and survive in the contaminated rhizosphere. MGEs also play essential roles to regulate the microbe–plant interactions. In *Pseudomonas fluorescens* Pf-5, it was found that many MGEs in the genome are related to the survival in the natural environment and the pathogenic relationships with plants [33]. Examples of the pathogenic genes include effector proteins and cell wall modification enzymes which aid the infection to plant [33]. HGT also plays important roles in the acquisition of genes related to bacterial functions that are beneficial to plants [34]. Examples of plant-beneficial functions performed by bacteria include phosphate solubilization, synthesis of antimicrobial compounds, inhibition of ethylene biosynthesis, induction of systemic disease resistance and synthesis of plant growth hormones [34]. Comparing among animal pathogens, phytopathogens, saprophytes, endophytes/symbionts and plant growth-promoting rhizobacteria (PGPRs), animal pathogens have the least genes related to plant-beneficial functions, while PGPRs have the most [34]. It is consistent with the fact that animal pathogens are the least related to plants while PGPRs are the most related to plants [34]. Twenty-three genes related to these plant-beneficial functions were selected for gene acquisition/loss analyses [34]. Most of them could be found in non-PGPRs and also in their distant ancestors. There is evolutionary evidence of transfers of these genes between different taxa. For example, 18 acquisition events of *nifHDK*, which is related to nitrogen fixation in proteobacterial PGPRs, and 21 acquisition events of *acdS*, which encodes 1-aminocyclopropane-1-carboxylate (ACC) deaminase, have been detected across phyla [34]. It was suggested that such ancient gene transfers resulted in the characteristic gene combinations in PGPRs [34].

In soil, biofilm attaches to soil particles, organic materials and other living organisms, including fungi and plant roots [35]. Biofilm, a consortium of bacteria enclosed in a self-produced matrix composed of polymeric molecules such as exopolysaccharides, proteins and nucleic acids, has been known to enhance HGT among bacteria, due to the increased physical contact among cells and the concentration of signaling chemicals and extra-cellular DNA in the biofilm matrix [36]. Recently, it was suggested that mycelia are the focal points for HGT among soil bacteria [37]. The network structures of mycelia were suggested to provide continuous liquid films to facilitate the movement of bacteria and thus promote the transfer of genetic materials among the bacteria [37].

### 3.2. HGTs among Filamentous Eukaryotes

In a survey of 659 published papers on 3617 species from 263 families of land plants, it was found that 80% of the species and 92% of the families are mycorrhizal [38]. Mycorrhizal plants interact with ectomycorrhizae or arbuscular mycorrhizae to form ectomycorrhizal or arbuscular mycorrhizal structures, with arbuscular mycorrhizae being the predominant and ancestral type of mycorrhizae [38]. Arbuscular mycorrhizae were also found to be able to alter the plant root microbial community in agricultural fields [39]. Therefore, arbuscular mycorrhizae are major players in the rhizosphere. The significance of arbuscular mycorrhizae in the rhizosphere has been extensively reviewed [40]. Although HGTs in fungi have been documented [18], due to the biased fungal genome information on the phylum *Ascomycota*, the transfers of genetic materials in other fungi, such as arbuscular mycorrhizae, are largely unknown. Some transfers of genetic elements among arbuscular mycorrhizae have been documented in satellite reports. Arbuscular mycorrhizae are coenocytic, in that different nucleotypes can coexist in the same cytoplasm [41,42]. Thus, segregation and genetic exchange during spore formation are possible [43]. Segregation refers to the random distribution of nuclei during spore formation, giving rise to offspring carrying different complements of nucleotypes compared to the parents and siblings [43]. On the other hand, genetic exchange refers to the mixing of nuclei of genetically different arbuscular mycorrhizae during spore formation to give rise to offspring having a mixture of parental nucleotypes [43,44]. MGE transfer among arbuscular mycorrhizae plays a role in the regulation of arbuscular mycorrhizae–plant interaction and plant growth. Using *Glomus intraradices* strains C2, C3 and D1 as the parental lines, it was shown that, with the different pairings, such genetic exchange between fungi could result in altered percentage colonization on the roots of *Oryza sativa* and *Plantago lanceolata* [45]. In *Oryza sativa*, it was shown that genetic exchange between strains of *Glomus intraradices* could either impair [43] or enhance plant growth [46], as well as affecting the speed of colonization by arbuscular mycorrhizal fungi (AMF) on the roots of *Oryza sativa* and the transcription of *Oryza sativa-*AMF symbiosis-related genes [46]. However, the interrelationships among the speed of colonization by AMF, effects on plant growth rate and alternations in gene expressions are still largely unknown [46].

By phylogenetic analyses, multiple HGTs from filamentous ascomycete fungi to oomycetes were inferred [47]. Later, it was found that such transfers resulted in the acquisition of genes related to various functions of oomycetes. The gene functions include the ability to invade plant cells, resistance to plant defense mechanisms and acquisition of nutrients and nucleic acids. It was then suggested that such transfers of genetic elements from fungi to oomycetes facilitated the evolution of oomycetes into effective plant parasites [48].

### 3.3. Cross-Kingdom HGTs

HGT has long been known as a phenomenon in prokaryotes. Later, inter-fungal HGTs were documented [18]. Recently, more and more evidence of the transfer of MGEs across different kingdoms is emerging. For example, in an evolutionary study of genes in nematode, it was found that a set of genes encoding cell wall-degrading enzymes, which are essential for successful parasitism of plants, were likely obtained from different bacteria through multiple independent HGT events [49], with possible gene duplications and gain of introns in the nematode genome after being transferred from bacteria [49]. Another example of cross-kingdom HGT is the transfer of *HhMAN1*, which encodes a mannanase for plant cell wall digestion, from bacterium to beetle (*Hypothenemus hampei*) [50]. *HhMAN1* was found in *H*. *hampei,* which attacks coffee plants, but not in the closely related species, *H. obscurus*, which is not a pest to coffee [50].

In a phylogenomic analysis of *Physcomitrella patens*, 128 genes from 57 families were found to be derived from prokaryotes, fungi or viruses [51], with many of them having been acquired from bacteria. One example is members of the subtilase gene family [51]. The finding of the transfer of subtilase genes from bacteria to *P. patens* agrees with the finding that subtilases in land plants are significantly different from those in fungi and animals [51,52]. Since subtilases play important roles in the development of lateral roots, cuticle and stomatal cells, it was suggested that the acquisition of subtilase genes by *P. patens* from bacteria facilitated the colonization of plants on land [51]. Other examples of genes acquired by *P. patens* from bacteria include those related to carbon metabolism, polyamine biosynthesis, auxin biosynthesis, purine degradation, nitrogen recycling, defense and stress tolerance [51]. Examples of these genes include the genes encoding acyl-activating enzyme 18 (AAE18) and YUCCA flavin monooxygenase (YUC3), which are involved in the biosynthesis of auxin [51]. Auxin regulates plant growth and development, including apical dominance and xylem differentiation [51]. Another example is *GCL,* which encodes glutamate-cysteine ligase for mediating plant disease resistance, photo-oxidative stress defense and heavy metal detoxification [51]. Homologs of *P. patens GCL* were only found in bacteria and green plants [51]. Two entry points of gene acquisition from bacteria to moss were proposed [51]. The first one was during spore germination and the early stage of gametophyte development of moss [51]. The second entry point was during fertilization and the early stage of embryo development of moss [51]. In bacteria, HGT has been regarded as a means to enable bacteria to obtain new characteristics for adapting to the environment [27]. Similarly, the transfer of genes from bacteria to moss, from which land plants evolved, suggests the role of HGT in the transition of plants from the aquatic to the terrestrial environment [51].

These examples show the importance of cross-kingdom HGTs in the adaptation of the recipients to the environment. Plants have also been documented as MGE donors. A phylogenomic analysis of *Oryza sativa* identified four plant-to-fungus HGT events [22]. It was suggested that such transfers could be either direct events or indirect transfers that involved prokaryotes [22]. Recently, it was reported that whitefly (*Bemisia tabaci*), an agricultural pest, obtained the gene *BtPMaT1*, which encodes phenolic glucoside malonyltransferase for neutralizing plant toxins, through HGT from plant [53]. Detailed examples of MGEs donated from plant in the rhizosphere will be discussed in Section 3.3.3.

Microbe–plant–microbe interactions in the rhizosphere have been extensively reviewed [54,55]. Plants exhibit mutually beneficial symbiotic relationships with various soil microbes [54,55]. Such proximity suggests the possibility of MGE transfers. The tripartite mutualism among bacteria, fungi and plants in the rhizosphere has been used as a model to illustrate certain microbe–plant interactions [56]. Examining the nature of cross-kingdom MGE transfers will provide information to comprehend such tripartite interactions and select beneficial microbial candidates among the interacting partners for agricultural use.

#### 3.3.1. Cross-Kingdom MGE Transfers among Microbes in the Rhizosphere Play an Important Role in Regulating Plant Growth and Development

By genomic sequence analyses, 65 genes homologous to *acdS*, which had been frequently transferred among bacteria [34], as discussed in Section 3.1, were found in filamentous eukaryotes including oomycetes and fungi [57]. ACC deaminase degrades ACC in root exudates and in turn inhibits the synthesis of ethylene, which is a plant growth inhibitor [58]. Soil microbes exhibiting ACC deaminase activities thus play an important role in promoting plant growth. The prokaryotic *acdS* genes and eukaryotic *acdS* genes share high amino acid and nucleotide sequence homologies [57]. Based on phylogenetic analyses, it was suggested that *Actinobacteria*, *Betaproteobacteria* and *Gammaproteobacteria* were the donors of *acdS* to filamentous eukaryotes [57]. However, the intron sequences among eukaryotic *acdS* were not conserved among different lineages of the filamentous eukaryotes [57]. Such a divergence in intron sequences is possibly a result of the evolution and domestication of each lineage after the gene acquisition [57]. Through genomic and transcriptomic analyses of the model arbuscular mycorrhiza, *Rhizophagus irregularis*, a class I ribonuclease III protein-coding gene was found to be transferred from cyanobacteria to *Glomeromycota*, the phylum to which *R. irregularis* belongs [59]. Plants are the hosts of AMF, which are hosts of *Burkhoderia*-related endobacteria (BRE) and *Mollicutes*/*Mycoplasma*-related endobacteria (MRE) [60]. By analyzing the genomes of *Diversispora epigaea* (formerly known as *Glomus versiforme*) and MRE, it was found that *D. epigaea* gained genes related to the bacterial methylation defense system from MRE, while the MRE gained genes related to fungal metabolism from *D. epigaea* [60]. It was suggested that such transfers of genetic materials between *D. epigaea* and MRE facilitated the adaptation of both organisms to their symbiosis [60].

#### 3.3.2. MGE Transfers from Rhizospheric Microbes to Plants

As discussed in Section 2, the transfer of MGEs from *Agrobacterium* to plant is a typical example of MGE transfer from microbes to plants. Besides bacteria, fungi are also donors of MGEs to plants. Through phylogenomic analyses of 1689 genes from 3177 families in *Oryza sativa*, 9 plant–fungi HGT events were identified [22]. Among them, five events were fungi-to-plant transfers while the other four were from plant to fungi [22]. The fungi-to-plant HGT events were supported by the branching of plant genes within the fungal cluster in the phylogenomic analysis [22], and include those genes involved in L-fucose uptake, siderophore biosynthesis, which is important for iron uptake and plant–fungal symbiosis, and those encoding membrane transporters and the phospholipase/carboxylesterase family of proteins [22]. Similar to the proposed significance of HGT from bacteria to plants [51], it was suggested that the plants gained functions beneficial for adapting to growing in soil via fungus-to-plant HGTs. HGTs from plants to fungi will be discussed in Section 3.3.3.

Recently, it was reported that nodulation of the host plant is modulated by rhizobial transfer RNA (tRNA)-derived small RNA fragments (tRFs) [6]. By examining soybean (*Glycine max* cultivar Williams 82) and rhizobium (*Bradyrhizobium japonicum* strain USDA110), 50 rhizobial tRFs were found in both the soybean nodules and the rhizobium culture. Twenty-five of these rhizobial tRFs were predicted to target fifty-two soybean genes, but not rhizobial genes [6], and they were not found in non-nodulated soybean tissues [6]. Soybean genes, *GmRHD3a/3b*, *GmHAM4a/4b* and *GmLRX5*, which are the predicted targets of the rhizobial tRFs, were found to have their mRNAs cleaved in the nodules but not in the uninoculated roots [6]. The mutation of *GmRHD3a/3b*, *GmHAM4a/4b* or *GmLRX5* led to the increased number of nodules when inoculated by rhizobium USDA110 [6]. This report demonstrated the cross-kingdom communication between bacterium and plant through tRFs and the promotion of soybean nodulation by rhizobial tRFs [6]. In the same report, rhizobial tFRs were also predicted to regulate soybean genes, which are annotated as encoding auxin receptors and efflux carriers, RING/U-box proteins and protein kinases [6]. These genes are all related to nodulation.

#### 3.3.3. Plants Could Be MGE Donors in the Rhizosphere Too

Plant-to-fungus HGT has been detected by phylogenomic analysis [22]. In another report, by genome analyses, 7 genes in *R. irregularis* were found to be obtained from bacteria, while 18 other genes were from bacteria or plants [23]. Three of these genes in *R. irregularis* were duplicated [23]. The expansion of the acquired genes in *R. irregularis* by gene duplication was also detected [23]. Examples are the plant-derived protein kinase gene family and the bacteria-derived cytotoxin gene family [23]. It was suggested that the acquisition of genes by *R. irregularis* assisted in its adaptation to symbiosis [23]. Besides DNA, RNA has also been found to be transferred from host plant to the infecting fungus. *Verticillium dahlia* is a fungal pathogen of various plant species. When the cotton plant was infected by *V*. *dahlia*, miR166 and miR159 were induced. Sequence analysis suggested that miR166 and miR159 are complementary to the fungal genes encoding Ca^2+^-dependent cysteine protease (Clp-1) and isotrichodermin C-15 hydroxylase (Hic-15), respectively. Clp-1 and Hic-15 were shown to be virulence factors [24]. MiR166 and miR159 were found in fungal hyphae isolated from the infected cotton plant, while *Clp-1* and *Hic-15* transcripts were reduced in the fungal hyphae [24]. Furthermore, it was shown that Arabidopsis could secrete exosome-like extracellular vesicles to deliver sRNAs. Upon infection by the pathogenic fungus, *Botrytis cinerea*, these vesicles accumulated at the infection sites and were then taken up by the fungus [25]. These sRNAs delivered from the infected plant to the fungal pathogen were found to silence fungal virulence genes [25]. These studies suggest plants could use mobile RNAs to defend against pathogens.

Examples of MGE transfers that regulate plant growth or microbe–plant interaction are summarized in Table 1.

## 4. The Effects of Stress and Environmental Factors on the Transfer of MGEs in the Rhizosphere

In the natural environment, organisms in the rhizosphere are constantly challenged by factors such as sub-optimal temperature and heavy metal contamination of soil [65]. Rhizospheric microbes such as PGPR can promote plant growth by enhancing the tolerance of plants to these stresses [66,67]. It has been shown that the transfer of MGEs can be affected by factors including soil temperature, soil depth and soil type [68]. In addition, other factors in the rhizosphere such as high nutrient availability can directly attract bacteria. Such attraction of bacteria towards plants provides more possibilities for MGE transfers between bacteria and plants [69].

### 4.1. Temperature

Temperature is an important environmental factor affecting MGE transfers. Between 10 and 35 °C, it was found that the MGE transfer frequency increased with air temperature [70]. Other studies using culture medium also support the importance of temperature on MGE transfer [71,72]. For example, the transfer of an antibiotic resistance gene was found to be at maximum efficiency at 25 °C [71]. Besides, it was demonstrated that a high temperature of approximately 41–45 °C can promote cell-to-cell plasmid transformation in *Escherichia coli*. Such transfers were more frequent in biofilm culture than in liquid culture [72]. These studies indicated that temperature is important for MGE transfer, and its efficiency in different bacteria varies with temperature, although most studies on the effects of temperature on the transfer of MGEs were limited to laboratory cultures.

### 4.2. Soil Composition

In a study addressing factors that affect the transfer of degradative plasmids in soil [70], compared to sand and sandy loam, loamy soil was found to be the most effective in increasing the frequency of plasmid transfers between bacteria. The recent trend of environmentally friendly agricultural practices promotes the use of organic fertilizers. It was shown that the use of either manure as organic fertilizer or inorganic fertilizer had different effects on the MGEs and microbial communities in the maize rhizosphere [73]. The use of an organic fertilizer significantly increased the relative abundance of MGEs in the rhizosphere. It was suggested that such an increase was related to the complex bacterial community in manure [73].

### 4.3. Soil Toxicity

As soil toxicity is becoming a serious global issue, research has been carried out to investigate the influence on soil microbes by soil toxicity. It was found that soil contaminated with antibiotics and pharmaceuticals had significant impacts on the rhizospheric microbiome and act as important hotspots for the transfer of MGEs [74]. A study demonstrated that the use of copper, considered an acceptable pesticide in organic agriculture, could inhibit the transfer of catabolic plasmids between microbial communities and could impede pesticide attenuation in the environment [75]. When cadmium (Cd), glyphosate and tetracycline were added to the soil where cassava was grown, plant growth was significantly inhibited, while the rhizospheric microbiomes were also dynamically changed [76]. In the rhizosphere, it was revealed by metagenomic sequencing that the glyphosate or tetracycline treatment increased the abundance of MGEs. A positive correlation between the concentration of glyphosate or tetracycline and MGE abundance was found. Such phenomenon suggested that the increased selective pressure from herbicides and antibiotics promotes the subsequent rapid evolution in the rhizospheric microbiome to adapt to the changing environment [76].

### 4.4. Influences by Plants

Planting has been suggested as a strategy to modulate soil properties. Furthermore, planting was demonstrated to enhance the transfer frequency of MGEs between bacteria in soil [70,77]. For example, the plantings of tomato, corn and wheat were shown to enhance plasmid transfer among soil bacteria [70]. Different plants exhibit different influences on the plasmid transfer frequency. Among tomato, corn and wheat, tomato was shown to exert the highest influence on the plasmid transfer frequency between bacteria in soil [70]. The contents of root exudates secreted by different plant species were suggested to be the reason behind such differences. Plant root exudates are known to be able to stimulate bacterial activities [78]. For tomato, organic acids identified in the root exudates were shown to induce the swarming mobility and chemotaxis response of bacteria [78]. It was also found that the plasmid transfer frequency was higher in the pea rhizosphere than in the barley rhizosphere. Root growth rate and distribution and exudate production were suggested as the reasons behind such a difference. It was shown that a certain range of phenol concentration in soil was beneficial for plasmid transfer between bacteria to some extent as well [70]. In another study, when the plant growth medium was treated with phenol, the planting of maize was shown to enhance plasmid transfers among the bacteria in the medium. The root exudates secreted by the maize plant were suggested to facilitate MGE transfer [79]. The use of catabolic MGEs for biological wastewater treatment in wastewater treatment plants (WWTPs) has been recently reviewed [80]. These results hint at the potential use of these MGEs for the remediation of contaminated soil.

In the rhizosphere, microbes can sense the surrounding environmental stimuli and modulate the process of gene exchange. For example, in legume symbiosis, rhizobia can transfer their symbiotic gene contents, called integrative and conjugative elements (ICEs), to other rhizobial genera after sensing certain plant compounds such as flavonoids. Such process can expand the host-range specificity of the rhizobium–plant interaction [64].

These evidences reveal the diverse stresses and environmental factors that can affect the transfer of MGEs in the rhizosphere. The transfers of MGEs among microbes in the rhizosphere can influence plant growth. On the other hand, the species of plants also influence the transfer of MGEs between microbes in the rhizosphere. These phenomena support the importance of microbiome–plant interaction in the rhizosphere on plant growth and soil remediation.

## 5. MGE Transfers Regulate the Adaptation to the Environment

MGE transfers among bacteria for spreading antibiotic and heavy metal resistance are typical examples of MGE-transfer-mediated adaptation to the environment [27,29,30,31,80]. Cross-kingdom MGE transfers also promote adaptation to the environment. For example, the ancient transfer of subtilase genes from bacteria to *P. patens* promoted the colonization of plants on the land [51]. Another example is the fungus-to-plant transfer of genes related to nutrient uptake from soil [22]. Such transfers promoted the growth of plants on soil. MGE transfers also play important roles in the adaptation to the dynamic composition of various species in the rhizosphere. For example, the transfer of MGEs between AMF and its symbiotic bacteria promoted their symbiosis [60]. The MGEs in rhizospheric microbes also regulate the pathogenic [33] or beneficial [34] relationship with plants. For example, microbial pathogenic genes carried by bacteriophages, plasmids and transposons include effector proteins and cell wall modification enzymes, which aid the infection to plant [33]. On the other hand, the *nifHDK* operon encodes nitrogenase in PGPR [34]. Nitrogenase converts atmospheric nitrogen into the ammoniacal form of nitrogen to be used by plants [62]. The *acdS* gene encodes ACC deaminase for promoting plant growth [34,57,63]. The ICE-mediated expansion of host range of rhizobia [64] is also potentially beneficial to plant growth. Compounds in plant root exudates are involved in the regulation of MGE transfer among bacteria. For example, in response to flavonoid in plant root exudate, rhizobia can transfer their symbiotic gene contents to other rhizobial genera [62]. The result is the expansion of the host-range specificity of the rhizobium–plant interaction [62]. MGE transfers allow the dynamic regulation of microbe–plant interaction in the rhizosphere, which is a constantly changing habitat.

## 6. Conclusions

The rhizosphere is a hotspot of MGE transfers among different species. Ancient intra-kingdom and cross-kingdom MGE transfers have been identified by phylogenetic analyses. It has been suggested that these ancient gene transfers helped to drive evolution and the adaptation of species to the environment they live in. Besides ancient MGE transfers that have been fixed in the genomes of the species involved, ongoing and recurrent MGE transfers during the interactions between species, such as those between plants and their associated soil microbes, have also been reported. These MGE transfers mediate pathogenic/beneficial microbe–plant interactions in the rhizosphere. The presence of multiple parties in the rhizosphere further complicates MGE transfer mechanisms. In the rhizosphere, bacteria, fungi and plants co-exist. MGE transfers happen between each of the parties and among all the parties. For example, the network structures of mycelia promote the transfer of MGEs among bacteria [37]. MGE transfers happen between fungus and its symbiotic bacterium [60]. MGE transfers also happen between plant and its symbiotic/pathogenic microbes [6,24]. Furthermore, root exudates from different plant species have different effects on MGE transfers among rhizospheric bacteria [70]. To make the story even more complicated, MGE transfers between microbes could influence microbe–plant interactions. For example, the transfer of *acdS* from bacterium to fungus is essential to the plant-growth-promoting role of the recipient fungus [57]. MGE transfers between bacteria could also influence bacterium–plant interactions. For example, the ICE transfer between rhizobia expands the plant host range of the recipient rhizobia [64]. The understanding of cross-kingdom MGE transfers in the rhizosphere helps us comprehend the tripartite mutualism among bacteria, fungi and plants. Moreover, the transfer of MGEs to mediate the degradation of toxins in the soil and the effects of environmental factors and root exudates on such transfers point to the potential use of microbe–plant systems for bioremediation. The transfers of MGEs in the rhizosphere and the factors affecting these transfers are illustrated in Figure 1.

## Figures and Tables

**Figure 1 biology-10-00477-f001:**
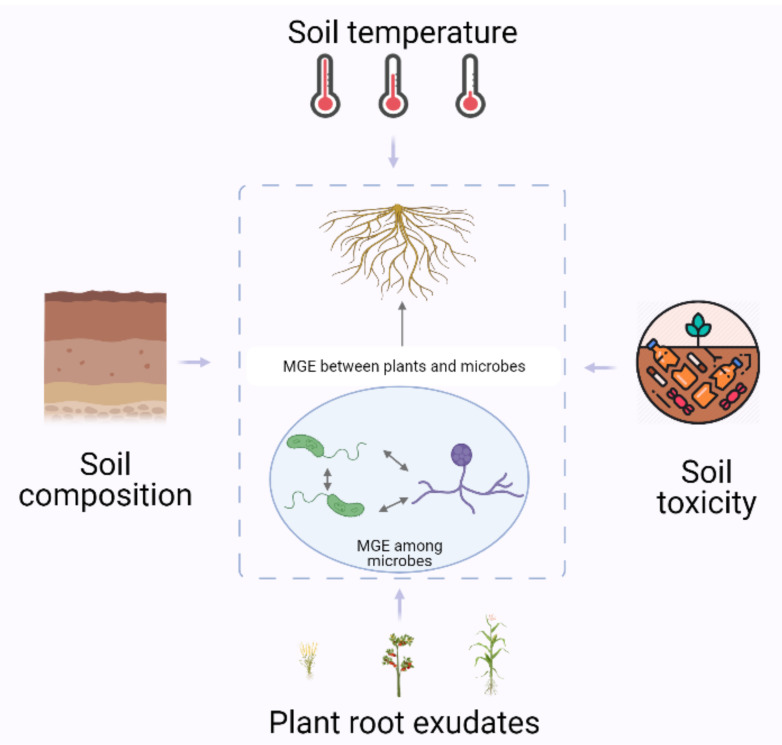
Mobile genetic elements (MGEs), including DNA and RNA, are extensively transferred among bacteria, fungi and plants in the rhizosphere. In the natural environment, factors including soil temperature, composition, toxicity and root exudates from plants influence MGE transfers in the rhizosphere. This figure was generated using BioRender.

**Table 1 biology-10-00477-t001:** Examples of MGE transfers that regulate plant growth or microbe–plant interaction.

MGE-Borne Gene(s)/MGE	Name of MGE-Borne Gene/MGE and Description	Nature	Direction	Significance	Reference
*nifHDK*	*nifHDK* is the operon that comprises the genes *nifH*, *nifD* and *nifK*. *nifH* and *nifD* encode the α subunit and the β subunit of dinitrogenase respectively, while *nifH* encodes the γ subunit of dinitrogenase reductase. Nitrogenase converts atmospheric nitrogen into the ammoniacal form of nitrogen to be used by plants.	DNA	Between bacteria	Encodes nitrogenase in PGPR; the transfer helped shape a taxonomic subgroup of PGPR	[34,61,62]
ACC deaminase structural gene	*acdS* encodes ACC deaminase which degrades ACC in root exudates and in turn inhibits the synthesis of ethylene. The result in the promotion of plant growth.	DNA	Between bacteria	The transfer helped shape a taxonomic subgroup of PGPR	[34,63]
			From bacterium to filamentous eukaryotes. including oomycetes and fungi	Promotes plant growth	[57]
Class I ribonuclease III protein-coding gene	*rirnc 2* encodes a class I ribonuclease III protein.	DNA	From cyanobacteria to *Glomeromycota*	Possible ancient symbiosis history between cyanobacteria and arbuscular mycorrhizae	[59]
Genes related to bacterial methylation defense system	Ribonuclease IIIs, Uma2 endonucleases, HNH endonuclease and methyltransferase. These genes are involved in the defense system against foreign DNA.	DNA	From *Mycoplasma*-related endobacteria to *Diversispora epigaea*	Facilitates the symbiosis of endobacteria and arbuscular mycorrhizae	[60]
Subtilase gene	It was suggested that genes of land plant subtilase family were derived from a single HGT event from bacteria. After that, rapid gene duplication occurred to give rise to the subtiliase family.	DNA	From bacterium to *P. patens*	Facilitates the colonization of plants on land	[51]
Gene for _L_-fucose uptake	*FucP*. FucP refers to _L_-fucose permease transporter family protein for _L_-fucose uptake.	DNA	From fungus to plant	Facilitates plant adaptation to growing in soil	[22]
Gene for membrane transporter	Protein sequence and domain analyses suggested that the gene belongs to the Major Facilitator Superfamily (MFS_1).	DNA	From fungus to plant	Facilitates plant adaptation to growing in soil	[22]
Gene for phospholipase/carboxylesterase family protein	Sequence analysis suggested that this gene has sequence similarity to the phospholipase/carboxylesterase protein family. Members of the phospholipase/carboxylesterase protein family have broad substrate specificity and the capacity of hydrolyzing carboxylic ester bonds.	DNA	From fungus to plant	Facilitates plant adaptation to growing in soil	[22]
Gene for siderophore biosynthesis	The gene for siderophore biosynthesis encodes a protein containing two domains: iucA and iucC. These two domains are involved in the sequential conversion of N epsilon-acetyl-N epsilon-hydroxylysine to the siderophore aerobactin.	DNA	From fungus to plant	Facilitates plant adaptation to growing in soil	[22]
Transfer RNA (tRNA)-derived small RNA fragments (tRFs)	Bj-tRF001, Bj-tRG002 and Bj-tRF003. Bj-tRF001, Bj-tRG002 and Bj-tRF003 target *GmRHD3a/3b*, *GmHAM4a/4b* and *GmLRX5* respectively. in soybean. The tRFs regulate nodulation of the soybean plant.	RNA	From rhizobium to soybean	These tRFs regulate nodulation related genes in soybean plant.	[6]
Cytotoxin gene family	Members of cytotoxin gene family encode proteins having conserved cytotoxic domians.	DNA	From bacterium to fungus	Promotes bacterium-fungus symbiosis	[23]
Protein kinase family	Members of protein kinase gene family encode proteins having conserved protein kinase domians.	DNA	From plant to fungus	Promotes fungus–plant symbiosis	[23]
miRNAs	miR166 and miR159. miR166 targets *Clp-1* transcripts, reduces virulence of pathogenic fungus. miR159 targets *Hic-15* transcripts, reduces virulence of pathogenic fungus	RNA	From plant to fungus	These miRNAs target fungal transcripts which are related to the fungal virulence.	[24]
sRNAs	TAS1c-siR483, TAS2-siR453 and IGN-siR1. TAS1c-siR483 target the *B. cinerea* genes BC1G_10728 and BC1G_10508. TAS2-siR453 targets the *B. cinerea* gene BC1G_08464. IGN-siR1 targets the *B. cinerea* gene BC1G_05327. These sRNAs target fungal genes which are related to the fungal virulence.	RNA	From plant to fungus	Silence fungal virulence genes, reduce virulence of pathogenic fungus.	[25]
Integrative and conjugative element (ICE)	ICE*^Ac^*. ICE*^Ac^* is able to excise, form a circular DNA and conjugatively transfer to the gly-tRNA gene of other rhizobial genera. Such integration expands the host range of the reciepent rhizobia.	DNA	Between bacteria	Expand the host-range specificity of rhizobia for interacting with plants	[64]

## Data Availability

Not applicable.

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
