# Peer review of "Rhizospheric Communication through Mobile Genetic Element Transfers for the Regulation of Microbe–Plant Interactions"

_biology, 2021, doi:10.3390/biology10060477_

Round 1

Reviewer 1 Report

I received this paper for review as a re-submission of previously rejected manuscript from another journal. I appreciate the changes in the paper made by authors during revision. Now, I am in favor to accept this manuscrip for publication. Moreover, the choice of the journal is now proper, and the scope of "Biology" corresponds much better to the topic of this review article than in the prevoius one. 

I have only minor comments to Table 1. The caption of the 1st column is not proper. particular genes are not MGEs. I suggest to use the title MGE-borne gene(s). Then, first two raws indicate names of particular genes, while other raws only describe what do particular gene(s) encode(s). I strongly encouragne the authors to provide names of all genes described in the table (they can be placed in parentheses, after description of their functions, if the authors think that mentioning these functions are necessary).

Author Response

Reviewer 1:

I received this paper for review as a re-submission of previously rejected manuscript from another journal. I appreciate the changes in the paper made by authors during revision. Now, I am in favor to accept this manuscript for publication. Moreover, the choice of the journal is now proper, and the scope of "Biology" corresponds much better to the topic of this review article than in the previous one.

I have only minor comments to Table 1. The caption of the 1st column is not proper. particular genes are not MGEs. I suggest to use the title MGE-borne gene(s). Then, first two rows indicate names of particular genes, while other rows only describe what do particular gene(s) encode(s). I strongly encourage the authors to provide names of all genes described in the table (they can be placed in parentheses, after description of their functions, if the authors think that mentioning these functions are necessary).

Response:

Thank you for approving the manuscript.

Table 1 has been amended accordingly. The caption of the first column is now changed to “MGE-borne gene(s)/ MGE”. “MGE-borne gene(s)” is indeed a more accurate description of the genes. Some tiny RNAs such as miRNAs and siRNAs are MGE itself. So, the term “MGE” is retained in the caption. The gene names and what the particular gene(s) encode(s) are shown in the second column captioned “Name of MGE-borne gene/ MGE and description”. The functions are also described in the second column.

Reviewer 2 Report

The manuscript is interesting and the revision is well-organized. Most of the relevant information on the topic is cited.

Minor suggestions:

1) Topic 5 is interesting and can be expanded specially in the part focusing the MGEs and the regulation of the pathogenic or beneficial relationship with plants (line 399).

2) Conclusion: "The presence of multiple parties in the rhizosphere further complicates MGE transfer mechanisms." Please elaborate on this.

Author Response

Reviewer 2:

The manuscript is interesting and the revision is well-organized. Most of the relevant information on the topic is cited.

Minor suggestions:

1) Topic 5 is interesting and can be expanded specially in the part focusing the MGEs and the regulation of the pathogenic or beneficial relationship with plants (line 399).

2) Conclusion: "The presence of multiple parties in the rhizosphere further complicates MGE transfer mechanisms." Please elaborate on this.

Response:

1) The part focusing the MGEs and the regulation of the pathogenic or beneficial relationship with plants has been expanded with the use of examples. (lines 405-411)

2) "The presence of multiple parties in the rhizosphere further complicates MGE transfer mechanisms." has been elaborated with the use of examples. (lines 426-437)